# The Olfactory Bulb in Companion Animals—Anatomy, Physiology, and Clinical Importance

**DOI:** 10.3390/brainsci13050713

**Published:** 2023-04-24

**Authors:** Rui Alvites, Abby Caine, Giunio Bruto Cherubini, Justina Prada, Artur Severo P. Varejão, Ana Colette Maurício

**Affiliations:** 1Centro de Estudos de Ciência Animal (CECA), Instituto de Ciências, Tecnologias e Agroambiente da Universidade do Porto (ICETA), Rua D. Manuel II, Apartado 55142, 4051-401 Porto, Portugal; ruialvites@hotmail.com; 2Departamento de Clínicas Veterinárias, Instituto de Ciências Biomédicas de Abel Salazar (ICBAS), Universidade do Porto (UP), Rua de Jorge Viterbo Ferreira, nº 228, 4050-313 Porto, Portugal; 3Associate Laboratory for Animal and Veterinary Science (AL4AnimalS), 1300-477 Lisboa, Portugal; jprada@utad.pt (J.P.); avarejao@utad.pt (A.S.P.V.); 4Instituto Universitário de Ciências da Saúde (CESPU), Avenida Central de Gandra 1317, 4585-116 Gandra, Portugal; 5Dick White Referrals, Station Farm, London Road, Six Mile Bottom, Cambridgeshire CB8 0UH, UK; abby.caine@linnaeusgroup.co.uk; 6Department of Veterinary Sciences, Veterinary Teaching Hospital “Mario Modenato”, University of Pisa, Via Livornese Lato Monte, San Piero a Grado, 56122 Pisa, Italy; giuniobruto@me.com; 7Centro de Ciência Animal e Veterinária (CECAV), Universidade de Trás-os-Montes e Alto Douro (UTAD), Quinta de Prados, 5001-801 Vila Real, Portugal; 8Departamento de Ciências Veterinárias, Universidade de Trás-os-Montes e Alto Douro (UTAD), Quinta de Prados, 5001-801 Vila Real, Portugal

**Keywords:** Olfactory Bulb, olfactory system, olfaction, olfactory anatomophysiology, companion animals, imagiology

## Abstract

The Olfactory Bulb is a component of the Olfactory System, in which it plays an essential role as an interface between the peripheral components and the cerebral cortex responsible for olfactory interpretation and discrimination. It is in this element that the first selective integration of olfactory stimuli occurs through a complex cell interaction that forwards the received olfactory information to higher cortical centers. Considering its position in the organizational hierarchy of the olfactory system, it is now known that changes in the Olfactory Bulb can lead to olfactory abnormalities. Through imaging techniques, it was possible to establish relationships between the occurrence of changes secondary to brain aging and senility, neurodegenerative diseases, head trauma, and infectious diseases with a decrease in the size of the Olfactory Bulb and in olfactory acuity. In companion animals, this relationship has also been identified, with observations of relations between the cranial conformation, the disposition, size, and shape of the Olfactory Bulb, and the occurrence of structural alterations associated with diseases with different etiologies. However, greater difficulty in quantitatively assessing olfactory acuity in animals and a manifestly smaller number of studies dedicated to this topic maintain a lack of concrete and unequivocal results in this field of veterinary sciences. The aim of this work is to revisit the Olfactory Bulb in companion animals in all its dimensions, review its anatomy and histological characteristics, physiological integration in the olfactory system, importance as a potential early indicator of the establishment of specific pathologies, as well as techniques of imaging evaluation for its in vivo clinical exploration.

## 1. Introduction

The olfactory system, as one of the five traditional sensory senses, is of fundamental importance in animal behavior and in its interaction with the surrounding environment. Its functions extend to roles as diverse as feeding and foraging, inter and intraspecific communication, exploration, and environmental recognition, predation or avoidance of predators, mating and reproduction, and territorial marking [1]. The capture, recognition, and processing of odorant molecules results from an interaction between different components of the olfactory system. The Olfactory Bulb (OB) is part of the main olfactory system and works as an intermediate component between the olfactory epithelium in the nasal cavity and the connections established with distinct parts of the brain. This component is a multilayered structure contained in the central nervous system that receives the information contained in the odoriferous stimuli captured by the olfactory receptors peripherally. These signals are transmitted in a hierarchical and highly organized way through complex interactions between primary and secondary neurons within the so-called olfactory glomeruli, and the properly filtered information is further transmitted to higher cortical levels [2]. The OB is one of the first structures to evolutionarily appear in the vertebrate brain [3], and its importance justifies the fact that, although neurogenesis in the adult brain registers mainly in the hippocampus and subventricular zone [4], the OB receives neuroblasts through the rostral migratory stream, allowing it to refine the animal’s sensitivity to smells [5].

The olfactory capacity is variable among different species, namely dependent on their behavioral and eating habits [6]. Predatory carnivores have a remarkable olfactory ability that is evolutionarily related to foraging and interaction with prey [7]. The dog has a recognized and superior olfactory acuity which proved to be essential in its evolutionary process [8] and that is not only used physiologically in the search and capture of prey, interaction with other animals and humans, and territory exploration and marking, but also in human dog-assisted activities as wide as identifying people, drugs, or illness [9].

The olfactory capacity in canids is considered to be about 10,000 to 100,000 times greater than that of humans, also presenting a much lower limit of detectability for odorant compounds [10]. When compared to humans, dogs are able to identify much lower concentrations of odorants due to an effective combination of a high density of olfactory neurons, specialized olfactory airflows, and specific central processing [9]. There are other intrinsic and individual factors that also seem to influence the olfactory capacity of each dog, such as the high genetic polymorphism associated with olfactory receptors and the consequent variability of olfactory acuity between different breeds [11]. Older animals tend to have reduced olfactory acuity due to degenerative phenomena observed in structures such as the olfactory epithelium, although they maintain better long-term memory for odor identification. Cells in the OBs in females also appear to be more active than those in males [12]. External and environmental factors can also interfere with the olfactory ability of dogs, with good levels of humidity, balanced temperatures, and even an adequate diet positively influencing olfactory detection [11,13]. The feline sense of smell is neither as explored nor as well understood as the canine one, but recent studies have created evidence that cats may have a capacity for olfactory discrimination far superior to that of dogs [14]. The difficulty in applying olfactory perception tests in cats, contrary to what happens in dogs [15], makes it difficult to unequivocally confirm this fact.

Despite the physiological and behavioral importance of olfaction in companion animals, only a few studies have focused on this system and there are limited data that unequivocally support the olfactory superiority of the dog, and even less of the cat. Among the components of the main olfactory system, those found downstream of the olfactory mucosa are the most difficult to explore, particularly those found intracranially such as the OB, which has been most widely studied structurally in rodents [16]. The inability to directly study the OB gains even greater relevance as it is realized that some clinical conditions and syndromes such as Canine Cognitive Dysfunction can translate into specific physical changes in this neural structure [17]. The only way to directly study the OB is its *post-mortem* histopathological analysis [18], with imaging techniques such as computed tomography (CT) and magnetic resonance imaging (MRI) being the traditional techniques to assess its shape and structure in vivo [19].

It is known that the OB, located in the rhinencephalon, acts as the interface between the external and internal olfactory systems, with both modulatory and sensory functions. This structure engages in the processing and initial filtering of olfactory information and allows detecting and discriminating odors and the transmission and integration of this information at the level of the olfactory cortex. It is an odor coding process based on a spatiotemporal pattern activation [20]. Recent findings have even allowed for establishing a direct relationship between the shape and dimensions of the OB and the olfactory function [21]. In companion animals, some advances have also been achieved in this subject, with some works allowing us to understand how variations in the dimensions and shape of the OB can influence the olfactory capacity of the animal, concluding that, in different breeds of dogs, the conformation of the head directly influences the angle and orientation of the OB and, consequently, the olfactory capacity [22,23]. There even seems to be a relationship between the morphological plasticity of the lamina cribrosa and the pressure induced by artificial selection in domestic canids compared to wild species [24]. Likewise, it was also noticed that species with greater olfactory acuity such as the dog have morphometric adaptations in the components of the olfactory system, including the OB, which differ from those observed in other species with lower olfactory performance such as herbivores [6]. In fact, evident morphometric differences are observed in structures such as the OB and the olfactory tracts that present dimensional reductions in humans and herbivores when compared to dogs, morphologically translating evolutionary and adaptive differences [25]. The cat has only been explored to a limited extent with regard to this topic.

The aim of this work is to revisit the OB in companion animals in all its dimensions, reviewing its anatomy and histological characteristics, physiological integration in the olfactory system, importance as a potential early indicator of the establishment of specific pathologies, and techniques of imaging evaluation for its in vivo clinical exploration.

## 2. Anatomy and Structural Organization of the Olfactory Bulb

The OB is a bulbar structure located in the rostral extremity of each cerebral hemisphere, ventrocranially to the frontal lobe [25] and near the orbital portion of the frontal bone, arranged on each of the ethmoidal fossae of the cribriform plate through which it receives the constituent filaments of the olfactory nerve. The two OBs are separated from each other by the crista galli of the ethmoid bone. In the dog it has a volume of around 1.8 mm^3^ in normosmic individuals, representing around 0.31% of the total brain volume; this percentage drops to 0.01% in the human brain. Integrated in the respective cerebral hemisphere, the OB represents around 43% of its total diameter against 9% in humans [25]. Although, in cats, there are no established reference values for the dimensions of the OB, it is accepted that felines have proportionally smaller OBs than carnivores [26]. It is also the first component of the basal portion of the rhinencephalon which, in turn, includes the structures of the basal telencephalon related to olfaction, the hippocampus, and associated structures [27].

Within the cranial cavity, the OB is located in the rostral cranial fossa, defined as the intracranial area extending from the optic canal and middle cranial fossa, caudally, and the cribriform plate, rostrally. The ethmoid fossa, separated from the nasal cavity by the cribriform plate, is an excavated continuation of the most rostral portion of the rostral cranial fossa. The rostral fossa receives not only the OB at the level of the ethmoid fossa, but also he olfactory peduncle and portions of the frontal lobe [28,29]. Internally, the OB can be filled by the so-called OB cavity or olfactory ventricle, an extension of the rostral horn of the lateral ventricles which, as such, is part of the cerebral ventricular system, allowing the free circulation of cerebrospinal fluid and participating in the rostral migratory stream [30,31].

Each OB is connected to the corresponding cerebral hemisphere through the olfactory peduncle, which has a short length and rapidly divides into the lateral, medial, and intermediate olfactory tracts or stria, each reaching a certain cortical territory [32]. The angle where the lateral and medial olfactory tracts separate rostrally delimits the so-called trigone or olfactory tubercle which, in turn, delimits the piriformis lobe cranially [25]. The rhinencephalon and, by extension, the OB and peduncles are separated from the medial and lateral portions of the corresponding hemispheres by the lateral and medial rhinal sulcus, which some authors also name as olfactory grooves, generating some inconsistency in the applied nomenclature. The lateral rhinal sulcus can be further divided into a rostral portion and a caudal portion [27]. The space that separates the OB from the rest of the brain was also tentatively referred to as the olfactory fissure [29].

## 3. Histology of the Olfactory Bulb

Histologically, the OB consists of seven layers. From the most superficial, these are: [1] Olfactory nerves layer; [2] Glomerular layer; [3] External plexiform layer; [4] Mitral cell layer; [5] Internal plexiform layer; [6] Granular cell layer; and [7] Subependymal or periventricular layer [18]. These layers are strictly organized, facilitating the processing of olfactory information and its spatial encoding (Figure 1 and Figure 2).

The olfactory nerve layer consists essentially of axons of olfactory neurons and glial cells that cross the lamina cribrosa from the olfactory mucosa. In the glomerular layer, several axons of the olfactory neurons are found in synapse with mitral cells, tufted cells, and interneurons in the olfactory glomeruli. The external plexiform layer contains the somata of tufted cells, the primary and secondary dendrites of mitral and tufted cells, and the branched secondary dendrites of granule cells. The mitral cell layer encompasses a cluster of conical mitral cell bodies aligned in a single row. The internal plexiform layer is thinner than the outer counterpart, lacks cell bodies, and contains only the axons of the mitral and tufted cells. The granular layer contains most of the cell bodies of the granule cells, and, finally, the subependymal layer is made up of the neurons that penetrate and leave the OB [18].

According to the corresponding cell layers, in the OB, it is possible to find three types of specific cells: mitral cells, tufted cells, and juxtaglomerular cells. The latter group includes at least three distinct types of cells—namely, periglomerular cells, external tufted cells, and the so-called superficial short-axon cells [33]. The continuous influx of new neurons that the OB receives through the rostral migratory stream allows their integration and renewal at the level of the granule or periglomerular layers, also allowing ongoing integration of new and different smells [5].

The mitral and tufted cells are both projection neurons whose axons communicate with the olfactory cortex. Mitral cells are the largest cells of the OB and also its largest efferent neurons. Its soma is in the mitral layer and its dendrites are classified as primary or secondary, and although both can be found in the outer plexiform layer, only the primary ones reach the olfactory glomerulus, one dendrite *per* glomerulus. The soma of the tufted cells lies in the outer plexiform layer and also projects a single primary dendritis towards a single glomerulus. In the glomeruli, the mitral and tufted cells are in synaptic contact with the olfactory neurons and also establish reciprocal synapsis with the periglomerular cells. Outside this structure, they only establish reciprocal synapses with the dendrites of granule cells through their secondary dendritis. Both mitral and tufted cells are efferent neurons of the OB, having crucial functions in the conduction and modulation of olfactory signals. Although the two types of cell share many morphological, functional, and molecular characteristics, there are obvious differences in the size of their soma, in the pattern of projection of their dendrites and axons, and even in the way in which they contribute to odorous responses [34].

Granule cells are present in the granular layer and are inhibitory interneurons with a small cell body. Their soma is located in the same granular layer, although they can also be in the internal plexiform layer and in the mitral cell layer. Their dendrites can branch in the outer plexiform layer, establishing reciprocal synapses with the secondary dendrites of the mitral and tufted cells. They are cells with axonless morphology, which means that their output occurs only through dendrodentritic synapses [35].

Juxtaglomerular cells are mostly interneurons that do not project outside the OB, and although they are traditionally considered together, they actually represent three morphologically heterogeneous cell types [33]. Periglomerular cells are the most abundant. They have a small cell body, normally project their dendrites only to one type of glomerulus and to a restricted area of it, and only rarely establish multiglomerular synapses. They may appear as axonless cells or present an axon that is located in the interglomerular space, projecting laterally. Like granule cells, they have inhibitory interneuron functions [36]. The outer tufted cells are those with the largest soma; their dendrites are mono- or di-glomerular and occupy a larger volume of the glomerulus. There are at least two types of these cells with variations regarding the presence of secondary dendrites and the extent of axonal projection. These cells are excitatory interneurons with important functions in glomerular processing and also in interglomerular coordination, in which case they may have inhibitory actions on the information originating from the contralateral bulb [33,34]. Finally, the superficial short-axon cells present a soma of intermediate size between the two previous types and have dendrites that project exclusively into the interglomerular space and axons that extend one or two glomeruli away. It is the smallest percentage of the juxtaglomerular cell population and its functions are still poorly understood, but they are thought to act as excitatory interneurons [33,37].

## 4. Olfactory Bulb Integrated in the Olfactory System

Olfaction involves two systems: the main olfactory system and the accessory olfactory system, also known as the vomeronasal system. The main olfactory system is concerned with detecting a large array of volatile odorants that translate into a consistent response known as a smell. In this system, essentially, the information contained in the compounds detected in the olfactory receptors of the olfactory epithelium, peripherally, is transmitted to the main integration centers, the OBs, and from there to higher cortical centers after adequate filtration and modulation [10,38]. In contrast, the vomeronasal system is activated by stimuli of reproductive and social origins, such as pheromones or kairomones, not triggering conscious responses after detection of the chemical molecule. This system involves the presence of a specialized sensory structure, the vomeronasal or Jacobson’s organ, and its interaction center is the accessory OB [39]. The vomeronasal organ is found between the nasal and oral cavities, close to the roof of the oral cavity, and the nasopalatine duct, which opens caudally to the upper incisors, allows a direct connection with the vomeronasal system and the arrival of pheromones—for example, through the flehmen reflex in some species [14,40]. Although some recent works have demonstrated that the two olfactory systems are essential for a correct interpretation of social and predation olfactory stimuli and are somewhat interrelated [41], physiologically, they are independent with regard to the reaction to captured volatile molecules and the cerebral pathways involved. Therefore, in the following segment, only the physiological sequence in which the OB is involved—that of the main olfactory system—is explored.

Olfactory perception begins when there is aspiration of air rich in volatile odorant molecules that reach the olfactory recess, a region covered by olfactory mucosa located in the caudo-dorsal region of the nasal cavity [42]. In all domestic mammals, this olfactory mucosa is easily recognizable in vivo due to its brownish-yellowish coloring, which allows it to be distinguished from the adjacent pink respiratory mucosa [25,40,43]. Airflow patterns through the nasal cavity in dogs are highly specialized, with the presence of a dorsal nasal meatus well separated from the respiratory airways and communicating directly with the well-developed olfactory recess [44]. This specialized separation of respiratory and olfactory air flows is also present in cats [40]. In the olfactory recess, the odorants connect to olfactory receptor proteins on the cilia of olfactory sensory neurons arranged on the surface of the olfactory neuroepithelium. In dogs, this neuroepithelium is found covering not only the lamina cribrosa but also the caudo-dorsal region of the nasal septum and portions of the nasoturbinates, maxilloturbinates, and ethmoturbinates [11,42], and can contain between 6 and 10 million neurons [45] which are first-order neurons. In felines, the olfactory epithelium appears more concentrated in the medial regions of the olfactory recess and with a reduced peripheral distribution, which translates into a smaller number of olfactory sensory neurons and, potentially, contributes to a lower olfactory capacity of these animals [26,40,46]. The olfactory neurons are bipolar neuronal cells whose round cell body is located in the middle stratum of the olfactory epithelium. The lifespan of olfactory neurons is only 30–60 days, with populations of stem cells located between the olfactory epithelium and the *lamina propria* continuously differentiating into new cells, both spontaneously and after injury [42]. In addition to the olfactory neurons, the olfactory epithelium also has sustentacular or supporting cells whose function is to provide structural support to active neurons and to phagocytize dead neurons and remaining odorants [47]. Additionally, there are Bowman’s glands responsible for the production of surface secretions that allow the dissolution of odorants, allowing their contact with the olfactory receptors, and protecting the neuroepithelium from the action of aggressive agents dragged by air flows to the olfactory recess [9]. Each olfactory neuron has a dendritic peripheral extension, single and short, which projects to the epithelial surface forming a protrusion called the olfactory vesicle from which a variable number of thin cilia depart radially and parallel to the epithelial surface, overlapping the cilia of neighboring cells and, thus, maximizing the odorant reception area. These cilia are covered by a phospholipid bilayer whose function is to expose different olfactory receptors present in its surface, connect and distinguish multiple odorant molecules, and initiate the olfactory nervous circuit [42]. Unlike humans, who may have around 50 million receptors [48], dogs can have more than 200 million olfactory receptors in the nasal cavity, maximizing their ability to identify a high number of odorants and in lower concentrations [11]. Each olfactory neuron expresses only one type of olfactory receptor, and the identification of an odor and its intensity depends on the number and combination of olfactory receptors activated simultaneously. This relationship is not, however, linear and there is a limited maximum allowable intensity which also depends on external factors such as concentration of odorants and exposure time [9]. Odorant recognition is dependent on G-protein-coupled receptors, whose extracellular portion is on the surface of the cilia and the intracellular portion is bound to a G-protein. When the odorant binds to the extracellular portion, it causes the breakdown of the A-subunit of the G-protein and the consequent activation of adenyl cyclase, converting ATP into cAMP. Through this mechanism, several sodium gated channels are activated, and the signal triggered by the odorant is amplified through depolarization. The respective action potential is transmitted downstream towards the central nervous system [49].

The basal expansion of olfactory neurons is a long, unmyelinated axonal process that leaves the olfactory epithelium. These axons cross the *lamina propria* as a set of axonal filaments surrounded by specific glial cells of the olfactory system, the ensheating cells, and it is the sum of all these components, the *fila olfactoria*, that constitutes the olfactory nerve itself [50]. These olfactory nerve bundles cross the cribriform plate of the ethmoid bone that separates the cranial cavity from the nasal cavity, also crossing the dura mater and arachnoid meninges while they are involved by the *pia mater*. Then, they penetrate the OBs located in the ethmoid fossa of the ethmoid bone, already in the telencephalon [42,51].

At the level of the OB, the first processing of olfactory signals generated in the olfactory neuroepithelium takes place. Here the axons of the olfactory neurons contact the dendrites of the mitral and tufted cells (each axon communicates with approximately 15 dendrites) and also with local interneurons, giving rise to a globular structure of synaptic units called the olfactory glomerulus. Each glomerulus is about 50 to 200 µm in diameter at a ratio of approximately 1000:1 axons per OB. The dog has about 5000 olfactory glomeruli in its OB, and each glomerulus results from the confluence of 5000 to 40,000 axons, thus observing a high afferent convergence ratio [52]. Comparatively, although in humans there may be a number of olfactory glomeruli identical to that identified in the dog, the convergence ratio is overwhelmingly lower, with values around 16:1 [53].

The OB has both sensory and modulatory functions through interactions between olfactory neurons, mitral and tuft cells, and different types of interneurons being involved in the filtration and processing of olfactory information, allowing odor differentiation and different levels of detection, and also the filtration of background odors [11]. Both the excitatory inputs and the intrabulbar circuits rigorously control the information that leaves the OB. This output is mainly controlled by periglomerular and granular cells, and the difference between the two is that the former mainly mediate interactions between cells affiliated with the same glomerulus, while granular cells will mediate interactions between neurons leaving different glomeruli. Thus, the propagation of action potentials and the excitatory message through the OB to the following olfactory pathways is controlled, guaranteeing a restriction on the information that goes to the subsequent levels [52].

From the olfactory glomeruli, the filtered and modulated olfactory impulses are transmitted by second-order neurons (mitral cells and tuft cells) to the primary cortical olfactory regions such as the anterior/rostral olfactory nuclei, which receive information from the OB and forward it through the lateral and medial olfactory stria to the olfactory tubercle, the piriform cortex, the entorhinal cortex, the septal region, and the peri-amygdaloid cortex. In addition to the ipsilateral network, it also has connections with the contralateral OB, anterior olfactory nucleus, and primary olfactory cortex, establishing a large number of interhemispheric projections [54]. In mammals, these connections allow odor information to be shared between the two hemispheres, ensuring interhemispheric perceptual unity and cortical isofunctional coordination [55]. Clinically, this allows, for example, that if there is a unilateral lesion in the structures upstream of the anterior olfactory nucleus, the information received in the contralateral components can reach both cerebral hemispheres, an evolutionary advantage in species physiologically dependent on smell, such as the dog or the cat. The olfactory cortex essentially allows for the identification and location of odors. Finally, third-order neurons make the connection between the primary olfactory areas and brain areas such as the orbitofrontal cortex and the hippocampus responsible, respectively, for the awareness of olfactory sensations and for memories associated with odors (Figure 3) [51].

From a functional point of view, the brain regions involved in olfaction can be divided into those that are part of the neocortex (orbitofrontal cortex) and are related to the awareness of smell, and those that are part of the limbic system. The limbic system, whose degree of complexity varies between distinct species of mammals, includes structures that together regulate the olfactory function but also its relationship with memory, motivation, reward mechanisms, and behavior. The limbic system includes amygdala (processing of smell-related fear and threat stimuli), entorhinal cortex (interface between hippocampus and cerebral cortex important for creating olfactory memories), and hippocampus (incorporation of long-term olfactory memories) [56]. There also seems to be a relationship between the limbic system and the isocortex, which is responsible for controlling higher-order functions such as conscious sensory perception cognition. While in humans there is an inverse relationship between the development of the isocortex and limbic system regions, carnivores have developed volumes of both regions, revealing a proportionately greater importance of olfaction in these species in the development of behavioral and cognitive components [11].

The majority of olfactory pathways are ipsilateral in their connections with perceptual areas in the brain and, recently, contrary to what was thought, it was noticed that the thalamus is involved in odor processing through the mediodorsal thalamic nucleus that receives information from olfactory areas such the amygdala, the piriform cortex, and the anterior olfactory nuclei and presents reciprocal connections with the orbitofrontal cortex [57]. The intermediate olfactory stria, through the rostral commissure, allows the connection between the OBs of both cerebral hemispheres, establishing an coordination between both bulbs, anterior olfactory nuclei, and piriform cortices [58]. In addition to the white matter tracts connecting the OB and the piriform cortex, the limbic system, and the entorhinal cortex, tracts connecting the OB and the occipital cortex, the posterior cingulate [59], the corticospinal tract, and the brainstem have also been identified. The connection with the brainstem allows the occurrence of olfactory-visceral phenomena such as salivation in response to odor stimuli. The connection between the OB and the occipital cortex is of particular importance in demonstrating the existence of an olfactory–visual integration that helps to understand the complex canine cognition and its interaction with the surrounding environment [32], The communication between the OB and different parts of the brain and of the central nervous system also allows us to understand the functional and unexpected consequences that changes in the bulb can have on the complex neural network.

## 5. Olfactory Bulb and Diseases

Although, in human medicine, the approach of the OB as a relevant element in clinical evaluation has already been established for some time, in veterinary medicine, its importance is still largely neglected in the fields of clinical research. In humans, there is a well-recognized correlation between olfactory alterations and other clinical entities, and the transition volumes between normal and hypoplastic OB are well defined [60]. It is considered that up to 20% of the general human population suffer from some type of olfactory dysfunction such as hyposmia (deficits in the sense of smell) or anosmia (total absence of sense of smell) [51]. In companion animals, the prevalence of hyposmia is largely ignored, mostly due to the difficulty in unequivocally and quantitatively assessing olfactory acuity in these species [8], but in animals highly dependent on the sense of smell for their physiology and interaction with the environment, even moderate changes in olfactory acuity can be serious. At the same time, it is known that several neurodegenerative disorders such as Alzheimer’s disease or Parkinson’s disease are associated with olfactory alterations that appear as prodromal symptoms [51]. Furthermore, the relationship between a partial olfactory loss and an increased risk of depression is also known, with this clinical state being associated with a lower activation of all the structures involved in olfaction perception and with a decrease in the dimensions of the OB [61].

The volume of the OB can effectively be used as a predictor of the individual’s olfactory function and of the response to a treatment for primary or secondary olfactory alterations, being proportional to the individual’s olfactory sensitivity and to the volume and density of gray matter in the primary olfactory regions. Scenarios of post-traumatic or post-infectious anosmia or hyposmia translate into smaller volumes of the OB compared to healthy individuals [62] and, in congenital anosmia, the OB may not be identifiable by MRI or may be hypoplastic. Conversely, an increase in olfactory acuity associated with larger OB volumes has also been identified [51,60,63]. In dogs, several infectious diseases such as canine distemper and canine parainfluenza, endocrine diseases such as hyperadrenocorticism, hypothyroidism, or diabetes mellitus and even nasal tumors have already been identified as affecting the olfactory capacity, but it is thought that in these cases the alterations are associated with changes within the nasal cavity and in the olfactory neuroepithelium. Other cases such as granulomatous meningoencephalitis or head trauma may have a direct influence on the brain regions related to olfactory processing, but it is not known at what level the OB may be involved [11].

In humans, olfactory function decreases with age due to various changes ranging from cumulative lesions in the olfactory epithelium, structural changes in the nasal cavity, and brain modifications secondary to the onset of neurodegenerative diseases [64]. Similar alterations can be identified in older dogs, which show both a significant decrease in the number of cilia and olfactory receptors in the olfactory neuroepithelium and senile changes in the OB such as cerebrovascular amyloidosis [65]. Some studies establish a relationship between the onset of neurodegenerative diseases and a decrease in OB volume in humans, with consequent impairment of the entire olfactory process [66,67]. Although brain structural changes have already been well identified in dogs suffering from cognitive disfunction [68], there is a lack of studies in companion animals that allow us to create a link between the decrease in OB volume and the establishment of such clinical scenarios. Hyposmia secondary to disease or trauma is normally reversible; the same not being observed in the case of changes associated with aging and neurodegenerative disorders.

Another common finding in older individuals and in certain pathologies, and which has already been identified both in humans and in dogs and cats, is the general enlargement of the ventricular system in normal patients and associated with disease processes [69]. In some individuals, the olfactory recess of the lateral ventricle, which is usually an empty virtual space, may become distended with cerebrospinal fluid (CSF). This OB ventricular dilatation has not been associated with ageing in geriatric cats [69,70] despite ageing being associated with generalized ventriculomegaly. In humans, generalized ventriculomegaly associated with an increase in the volume and pressure of the CSF and consequent compression of the adjacent brain tissue is often secondary to changes in fluid drainage associated with a variety of conditions including uncontrolled diabetes mellitus, [71], or as an *ex-vacuo* effect due to atrophy and loss of neuronal tissue in senility [72]. Currently, there are no studies suggesting that dilation of the olfactory recesses in cats and dogs are markers for senility or systemic disease, although pathologic obstruction of ventricular drainage in cases with forebrain neoplasia has been described as causing dilation of the olfactory recesses in companion animals [58].

## 6. Imaging Assessment of the Olfactory Bulb

Assessing the surrounding bony structures—namely, the lamina cribrosa—can be the best approach to the OB through CT [73], which shows a thin mineral-attenuating curved plate with small perforations visible on high resolution reconstructions (Figure 4). Determining the detailed volume and shape of a structure as small as the OB itself should favor the use of high-resolution MRI. Since the OB is positioned within the olfactory fossa of the ethmoid bone, there is a risk of susceptibility artifacts which lead to regions of artifactual signal void. Susceptibility artifact arises here from distortions in the magnetic field homogeneity, particularly due to the proximity of paramagnetic air in the adjacent frontal sinuses. Patients with smaller frontal sinuses (such as brachycephalic breeds) are affected to a lesser extent than small patients with relatively large frontal sinuses, such as cats [74]. In clinical practice, the use of T2-weighted sequences is common as the contrast between the OB tissue and the surrounding CSF allows the structure to be better defined [51].

The OB in the dog can be examined in the sagittal plane, showing uniform grey matter intensity brain parenchyma resting in the olfactory fossa and circumscribed rostrally by the cribriform plate. The olfactory fissure separates it from the rest of the brain with a thin band of CSF (hyperintense on T2W, hypointense on T1W) [29]. The bulb has a different orientation in different skull conformations, lying rostral to the frontal lobe in mesaticephalic or dolicocephalic breeds, and ventral to it in brachycephalic breeds (Figure 5).

After identifying the structure, the OB, its anatomic variations in terms of shape, volume, angle, and orientation relative to the rest of the brain have been studied. The correct quantification of the OB volume through MRI can be complex, and there are methods of volumetric determination by coronal T2-weighted segmentation applied in human medicine [75], although the use of these techniques in routine clinical practice is still limited. The shape of the OB appears to be a direct consequence of its adaptation to a greater or lesser depth of the lamina cribrosa in different breeds [24]. The OB angle is determined by measuring the angle formed between the olfactory fissure and the line joining the oral aspect of the hard palate and the intercondylar notch of the foramen magnum (baseline of the cranial cavity) on the sagittal image [22,23]. The orientation of the OB can be categorized from 1 to 5, with a higher number assigned to animals with a more ventral orientation of the OB [23]. Brachycephalic dogs have a smaller OB angle and, therefore, a ventral orientation of the OB in an adaptation secondary to the morphological changes in the cranial conformation in these breeds with, conversely, a larger OB angle in mesaticephalic and dolichocephalic dogs [23]. These data on OB angle have been used to quantitatively compare the degree of brachycephaly and the development of changes related to this conformation such as brachycephalic obstructive airway syndrome (BOAS) with a correlation to soft palate thickness and other alterations in the central nervous system.

Anatomic variations in the OB may also be related to the animal’s olfactory capacity, which physiologically seems to be linked to the cranial morphology, including the conformation of the lamina cribrosa and the shape and size of the OB that occupies the space over the olfactory fossa [76]. While some studies indicate otherwise [77,78], it is accepted that the brachycephalic breeds have an olfactory acuity and discrimination lower than the other breeds. Although the changes in the nasal cavity itself and the consequent hindering in odorant supply also contribute to a reduced olfaction, the reduction of brain regions related to olfactory discrimination due to compression of the OB and brain in general seem to contribute greatly to this lower olfactory performance [76].

Enlargement of the ventricular system—in particular of the olfactory recess of the lateral ventricle, associated with an atrophy and compression of the adjacent neural tissue— may be secondary to the impairment of normal CSF flow in dogs and, therefore, be associated with general neurological signs [69,79]. Eventually it may cause atrophy and compression of adjacent neural tissue and is potentially related to olfactory impairment. T2W MRI is the method of choice to evaluate for the presence of an olfactory ventricle, as the high signal of CSF on T2W identifies the recess as distinct from the brain parenchyma.

Regarding g the cat, there are fewer studies assessing the range of anatomic variation of the OB seen on imaging. Most cats have the OB located rostral to the brain, similar to dolichocephalic dogs. Some brachycephalic cats have a less protuberant OB, with breeds such as Persians described as having shortened OBs that are ventrally and laterally displaced [80], similar to brachycephalic dogs. Dilation of the olfactory recess of the lateral ventricle is identified as an incidental finding in normal cats [70]. In this study, 11% of cats with olfactory recess dilation were clinically normal and, therefore, this dilation was considered an incidental discovery in the absence of other MRI findings. There was, however, a greater likelihood of having bilateral dilation if it was secondary to ventriculomegaly or an intracranial space-occupying lesion. The normal appearance on MRI of the OB in a cat with normal OBs and of a cat with a dilated olfactory recess but presumptively clinically normal are included in Figure 6.

The morphology of the OB may be studied with MRI in both dogs and cats. In addition to identifying pathologies that may affect the OB, MRI can be an useful research tool, with OB angle used as a marker for brachycephaly allowing correlation with a variety of diseases such as BOAS, and also as a research tool that may indirectly assess the potential olfactory acuity of the dog.

## 7. Conclusions

The physiological particularities of the OB still raise doubts and questions. Its function as the first filtration and neuromodulation site for olfactory information coming from the olfactory neuroepithelium demonstrates its importance in olfactory perception, essential for a large number of organic functions ranging from feeding, inter and intraspecific interaction, and environmental exploration. Likewise, the fact that the OB is a continuous receiver of new neuroblasts coming from the subventricular zone through the rostral migratory stream demonstrates the importance that correct neuronal functioning and renewal have in this structure. The neuropathways established between the OB and different regions of the brain, like the so-called olfactory cortex and areas such as the orbitofrontal cortex, brainstem, and cortices associated with other sensory systems, not only reinforce the importance of the olfaction and its components in cognition and homeostasis, but also show that changes in this system can lead to unexpected deviations in the neurological field.

Clinically, efforts have been made to understand how morphological and dimensional changes in the OB can be early indicators of the presence or development of pathologies affecting the brain in general and the olfactory system in particular. In humans, reduced dimensions of the OB have already been related to observation of brain alterations associated with senility and neurodegenerative diseases, head trauma, and infectious diseases where the decreases in olfactory capacity are known consequences. The inverse relationship also exists. In companion animals, the inter-individual variability associated with OB is even greater, with breeds with different cranial conformations showing great variations in the disposition, dimensions, and shape of the bulbs, which translate into marked differences in the olfactory capacity of each animal. These findings, mostly identified by imaging, open doors to the use of OB as a signage element within the central nervous system that may be used as an early indicator of the establishment of neurodegenerative diseases associated with aging; as an element of prediction for the potential olfactory capacity of a young animal; and even for the probability of an animal to develop complications associated with brachycephalic syndrome based on the correlations established between the OB and the soft palate.

If this type of study is abundant in Human Medicine, the number is significantly reduced in companion animals, where the olfactory system and its components are often given little attention, the use of MRI as an evaluation method is not always available routinely, and the cat has been particularly underexplored. More in-depth studies are needed to reverse the doubts that persist on this topic.

## Figures and Tables

**Figure 1 brainsci-13-00713-f001:**
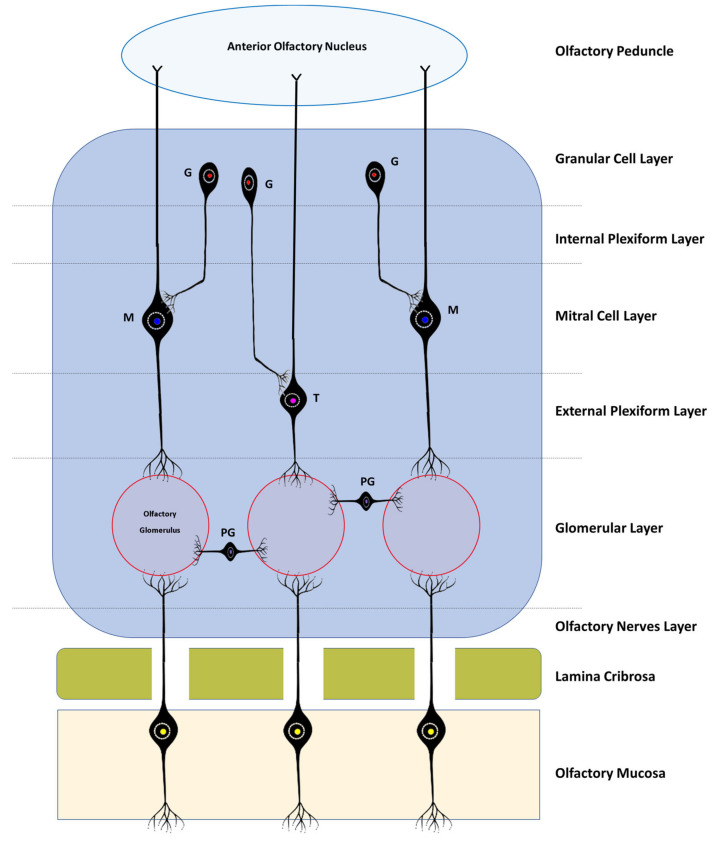
Simplified schematic representation of the histological layers of the OB, the interactions between the different bulbar cells, and the basic neural circuits in the OB. All the axons of the olfactory sensory cells project to a certain olfactory glomerulus depending on the type of receptor they have; PG—Periglomerular Cells; T—Tufted Cells; M—Mitral Cells; G—Granule cells.

**Figure 2 brainsci-13-00713-f002:**
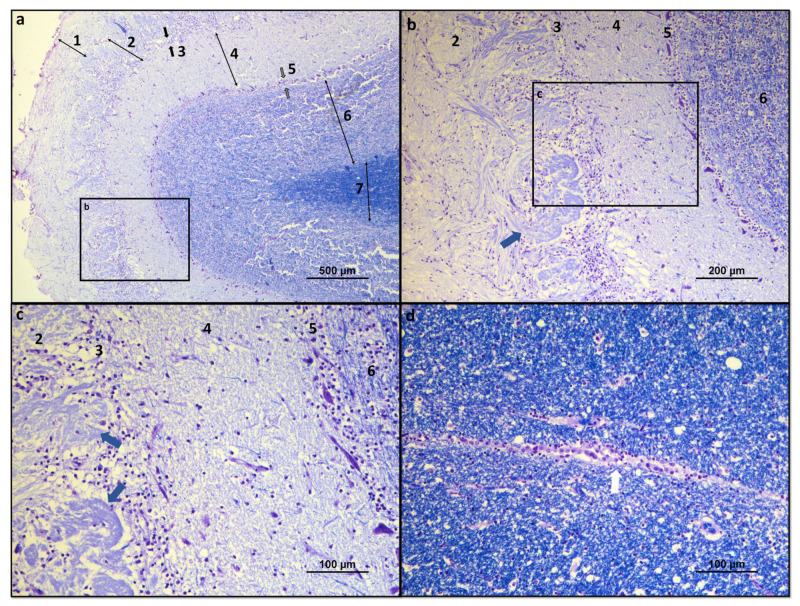
Histological overview of the dog OB: (**a**) Histological organization of the seven constituent layers of the olfactory OB: **1**—olfactory nerves layer; **2**—glomerular layer; **3**—external plexiform layer (between black arrows); **4**—mitral cell layer; **5**—internal plexiform layer (between gray arrows); **6**—granular cell layer; and **7**—subependymal layer. (**b**) Structural details of the glomerular, external plexiform, mitral, internal plexiform, and granular layers. Olfactory glomerulus highlighted by blue arrow; (**c**) Structural details of the glomerular, external plexiform, mitral, internal plexiform, and granular layers. Olfactory glomeruli evidenced by blue arrows; (**d**) Structural details of the ependymal layer of the OB surrounding the olfactory ventricle, highlighted by a white arrow. Luxol fast blue and cresyl violet.

**Figure 3 brainsci-13-00713-f003:**
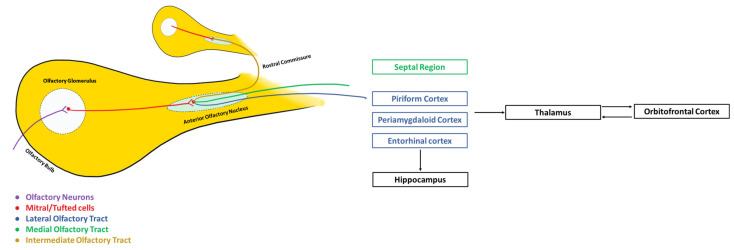
Simplified schematic representation of the olfactory pathways and their components.

**Figure 4 brainsci-13-00713-f004:**
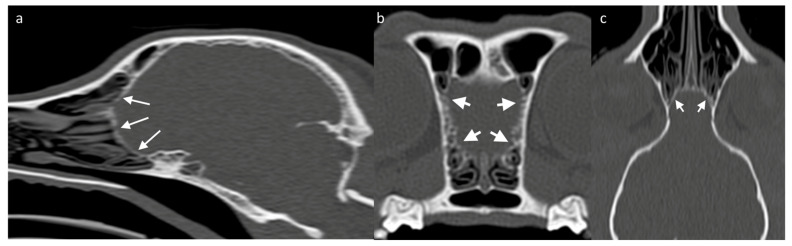
CT skull of a 3-year-old Jack Russell Terrier, acquired as a volume and reconstructed in a high frequency reconstruction algorithm, displayed in a bone window. (**a**) Sagittal/slightly parasagittal reconstruction to line through with the OB; (**b**) Transverse plane at the level of the OB; and (**c**) Dorsal plane reconstruction at the level of the OBs. White arrows on each image highlight the thin curved lamina cribrosa (cribriform plate) that bounds the rostral aspect of the OBs. Multiple tiny defects through which the olfactory nerve bundles pass are visible in all planes, although most distinct on the transverse plane compared to the remaining ones.

**Figure 5 brainsci-13-00713-f005:**
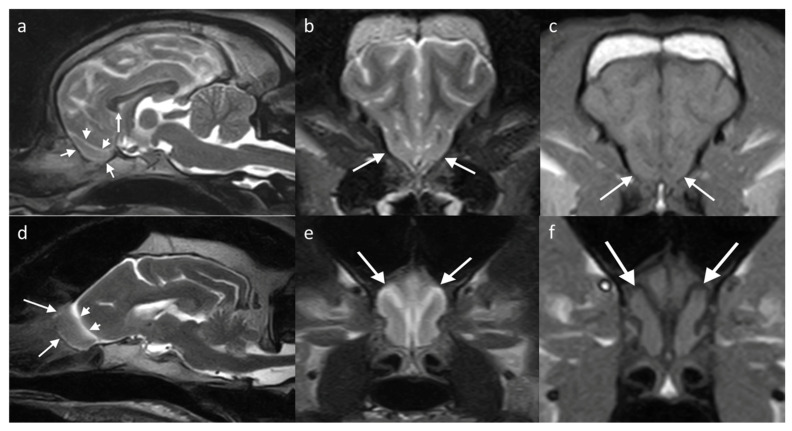
MRI brain of two dogs with different conformation, showing the location of the OBs (white arrows show the rostral extent, bounded by the lamina cribrosa (cribriform plate)). (**a**,**d**) T2W sagittal/slightly parasagittal to line through with the OB, with white arrowheads highlighting the olfactory fissure which separates the OB from the frontal lobe; (**b**,**e**) T2W transverse at the level of the OB; and (**c**,**f**) T1W transverse equivalent. (**a**–**c**) are of a brachycephalic breed (6-year-old French Bulldog) showing that the OBs are located on the rostroventral aspect of the brain; therefore, are seen ventral to the frontal lobes on the transverse view. (**d**–**f**) is a mesocephalic breed (12-year-old Golden retriever) with the OBs protruding from the rostral aspect of the brain. Transverse slices from patients with this conformation therefore show only the OBs and small segments of the surrounding brain.

**Figure 6 brainsci-13-00713-f006:**
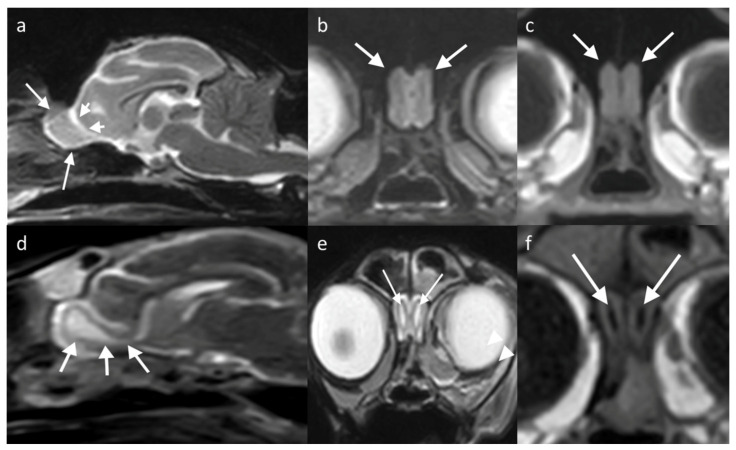
MRI brain of two cats, showing the location and appearance of the OBs (white arrows). (**a**,**d**) T2W sagittal/slightly parasagittal to line through with the OB; (**b**,**e**) T2W transverse at the level of the OB; and (**c**,**f**) T1W transverse equivalent. (**a**–**c**) is of a domestic short hair cat (8-year-old) showing that the OBs are located protruding from the rostral aspect of the brain, extending slightly more rostrally than in dogs. On the transverse view, the OBs are noted between the globes. (**d**–**f**) is of a 16-year-old cat with a common variation of anatomy, showing a distension of the olfactory recess of the lateral ventricles, leading to a T2W hyperintense and T1W hypointense central ovoid structure within the OB, which connects to the lateral ventricle via a thin stalk (white arrowheads).

## Data Availability

The data that support the findings of this study are available from the corresponding author on request.

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
