# Peer review of "The Olfactory Bulb in Companion Animals—Anatomy, Physiology, and Clinical Importance"

_brainsci, 2023, doi:10.3390/brainsci13050713_

Round 1

Reviewer 1 Report

Please check for the revisions below. Some major and minor revisions required.

The authors in the review explain the anatomy of the first central processing region, that is the olfactory bulb, its function, and the processing of olfactory information in companion animals, majorly dogs. Overall, the authors summed up the structural and functional organization as well as the role of OB in veterinary medicine. However, I have some points of concern to raise:

1.       In section 2: Anatomy and structural organization of the OB:

I expect anatomical differences between dogs, humans, and cats (if any literature exist) to be specified for the readers to better understand the purpose of coming up with a separate publication for companion animals. In line 132, cut off volume of OB has been defined for dogs; however, no similar work has been explained either in humans or cats (some references).

2.       Anterior olfactory nucleus is another region which is well defined in rats. It receives olfactory input from OB and projects it to other primary olfactory regions. I would look for the role of anterior olfactory nucleus in targeted animals. It would be nice to project some work done.

3.       In line 302, I would expect the authors to also mention the olfactory glomeruli in humans for the readers to better connect with the differences. Just an overview would be ok, not much in detail.

4.       In line 331, citing reference 43, authors defined the limbic system. The study cited was done in rats, does the citing refer to dogs or humans here?  Also, I wonder why OB is included in the limbic system?

5.       I found most of the references cited are for rodents. Yet, authors cite this for animals, which is okay to prove a point. But it makes it harder for the reader to follow. Are there no studies defining such work in animals? A reference to humans would also be something of interest for the wider population.

6.       Are there any behavioral studies conducted that help in clinical diagnosis of animals? If so, please specify.

Minor-

1.       I would suggest assessment of English language as there exist some spelling mistakes.

2.       Please stick to the abbreviation once defined (OB).

Manuscript needs to be corrected for native English.

Author Response

Answer to reviewer 1

Dear reviewer 1:

Thank you very much for the feedback on this review phase, and also for the suggestions made, which received the best attention from us. The authors inform that the final document has been revised and all suggestions made by the reviewers have been introduced, being duly identified in the final document with highlight in different colors. This review also made it possible to identify and correct some errors and typos as well as improve the general level of English.

The changes made to the document are described below. All changes introduced and highlighted text segments appear in the final document highlighted in green.

  • In section 2: Anatomy and structural organization of the OB: I expect anatomical differences between dogs, humans, and cats (if any literature exist) to be specified for the readers to better understand the purpose of coming up with a separate publication for companion animals. In line 132, cut off volume of OB has been defined for dogs; however, no similar work has been explained either in humans or cats(some references).

The textual segment has been modified and new information has been introduced, appearing highlighted in green:

“In the dog it has a volume around 1.8 mm3 in normosmic individuals, representing around 0.31% of the total brain volume, this percentage dropping to 0.01% in the human brain. Integrated in the respective cerebral hemisphere, the OB represents around 43% of its  total diameter, against 9% in humans (25). Although in cats there are no established reference values for the dimensions of the OB, it is accepted that felines have proportionally smaller OBs than carnivores (26).”

  • Anterior olfactory nucleus is another region which is well defined in rats. It receives olfactory input from OB and projects it to other primary olfactory regions. I would look for the role of anterior olfactory nucleus in targeted animals. It would be nice to project some work done.

A new textual segment has been added for further information on the importance of the anterior olfactory nucleus, highlighted in green:

In addition to the ipsilateral network, it also has connections with the contralateral OB, anterior olfactory nucleus, and primary olfactory cortex, establishing a large number of interhemispheric projections (54). In mammals, these connections allow odor information to be shared between the two hemispheres, ensuring interhemispheric perceptual unity and cortical isofunctional coordination (55). Clinically, this allows, for example, that if there is a unilateral lesion in the structures upstream of the anterior olfactory nucleus, the information received in the contralateral components can reach both cerebral hemispheres, an evolutionary advantage in species physiologically dependent on smell, such as the dog or the cat.”

  • In line 302, I would expect the authors to also mention the olfactory glomeruli in humans for the readers to better connect with the differences. Justan overview would be ok, not much in detail.

A new textual segment and reference was added to integrate the new information, highlighted in green:

“Comparatively, although in humans there may be a number of olfactory glomeruli identical to that identified in the dog, the convergence ratio is overwhelmingly lower, with values around 16:1 (53).”

  • In line 331, citing reference 43, authors defined the limbic system. The study cited was done in rats, does the citing refer to dogs or humans here?

Also, I wonder why OB is included in the limbic system?

A reference was added concerning the dog, as well as a new textual segment with new information about the limbic system in these species compared to humans, highlighted in green. The inclusion of the olfactory bulb in the limbic system was an error and has been corrected.

“There also seems to be a relationship between the limbic system and the isocortex, which is responsible for controlling higher-order functions such as conscious sensory perception cognition. While in humans there is an inverse relationship between the development of the isocortex and limbic system regions, carnivores have developed volumes of both regions, revealing a proportionately greater importance of olfaction in these species in the development of behavioral and cognitive components (11).”

  • I found most of the references cited are for rodents. Yet, authors cite this for animals, which is okay to prove a point. But it makes it harder for the reader to follow. Are there no studies defining such work in animals? A reference to humans would also be something of interest for the wider population.

The authors agree that a significant part of the description the olfactory system are based on information from rodents. This is due precisely to the fact that these species are much more explored and are better characterized from the histological and anatomical point of view than companion animals, which makes finding unequivocal information about these species more difficult. Even so, the authors considered that the information based on rodents do not vary greatly from those extrapolated to carnivores, allowing the reader to perceive the complexity of the anatomophysiological organization in these species, with the appropriate adaptations. Furthermore, in addition to the references added, related to the description and characterization of structures in humans and companion animals as requested by the reviewer, a textual segment was added and complexified on the anatomy and histology of the nasal cavity of dogs and cats, what works as an answer to the weak point indicated by the reviewer. The same segment appears in the text in blue, as it was introduced following a request also made by another reviewer.

“In all domestic mammals, this olfactory mucosa is easily recognizable in vivo due to its brownish-yellowish coloring, which allows it to be distinguished from the adjacent pink respiratory mucosa (25, 40, 43). Airflow patterns through the nasal cavity in dogs are highly specialized, with the presence of a dorsal nasal meatus well separated from the respiratory airways and communicating directly with the well-developed olfactory recess. (44). This specialized separation of respiratory and olfactory air flows is also present in cats (40). In the olfactory recess, the odorants connect to olfactory receptor proteins on the cilia of olfactory sensory neurons arranged on the surface of the olfactory neuroepithelium. In dogs, this neuroepithelium is found covering not only the lamina cribrosa but also the caudodorsal region of the nasal septum, portions of the nasoturbinates and maxillo-turbinates and of the ethmoturbinates (11, 42), and can contain between 6 and 10 million neurons (45) which are first-order neurons. In felines, the olfactory epithelium appears more concentrated in the medial regions of the olfactory recess and with a reduced peripheral distribution, which translates into a smaller number of olfactory sensory neurons and potentially contributes to a lower olfactory capacity of these animals (26, 40, 46).”

  • Are there any behavioral studies conducted that help in clinical diagnosis of animals? If so, please specify.

The olfactory system and the consequences of its changes are repeatedly undervalued in the clinical practice of veterinary medicine. The neurological examination traditionally applied in the evaluation of the nervous system and the cranial nerves in companion animals is very sparse in approaching the olfactory nerve, and its evaluation is often restricted to an attempt to understand whether the animal can smell an odor or not ( with techniques as fallible as covering the animal's eyes and exposing the nostril to food, without considering the complexity of the olfactory system and its physiological relationships). The canine sense of smell turns out to be much more valued when the animal has a "profession" in which this ability is used, for example in trail dogs, drug detection dogs, police or hunter dogs and more recently dogs capable of detecting diseases. Even so, in these cases, the animals are mostly evaluated for their ability to fulfill the mission and not so much for variations in their olfactory ability. Some authors have tried to develop reliable tests that allow the evaluation of olfactory discrimination compared between animals, but once again based exclusively on the ability to identify odors without considering other physiological surroundings (ex: Polgár, Z., Kinnunen, M., Újváry, D., Miklósi, Á., & Gácsi, M. (2016). A test of canine olfactory capacity: comparing various dog breeds and wolves in a natural detection task. PloS one, 11(5), e0154087.). This limitation is precisely the one that the authors intend to overcome, and this article appears as an introduction to more complex works in the fields of animal behavior and imaging that we intend to develop in this field of veterinary medicine.

  • I would suggest assessment of English language as there exist some spelling mistakes.

A careful review was carried out throughout the document to correct minor errors identified and improve English.

  • Please stick to the abbreviation once defined (OB).

A careful review was carried out throughout the document to ensure correct use of defined abbreviations.

Reviewer 2 Report

Alvites et al. review the literature on the olfactory bulb, a key central nervous relay in olfactory pathways, in domestic animals. 

This discourse consists mostly of long passages that belong better in textbooks. Most of the details have nothing directly to do with "companion animals".

The task of a good review paper should be more in its entirety than the sum of the individual reports cited from the literature.

Apart from 2 paragraphs at the end of the review, there is no significant new message in this report.

Minor:

Abstract: 

·      line 1: delete “main” (OB also plays a part in VNO in some animals)

·      what are “neurogenerative diseases”?

The introduction contains many general well-known statements of trivial importance, which are in part disputable (e.g., the notion that dogs smell odorants at a much lower concentration than humans. This is strictly odorant-dependent and does not apply for all odorants (see https://pubmed.ncbi.nlm.nih.gov/24278296/)

Thus, the term “microsmatic or macrosmatic”), as often used in popular scientific publications, should be, if not avoided, at least explained.

There are some unnecessary repetitions, e.g., the location of the OB

Citations on the structure/function of neuroepithelium most often derive from mice and do not compare the situation in companion animals.

Author Response

Answer to reviewer 2

Dear reviewer 2:

Thank you very much for the feedback on this review phase, and also for the suggestions made, which received the best attention from us. The authors inform that the final document has been revised and all suggestions made by the reviewers have been introduced, being duly identified in the final document with highlight in different colors. This review also made it possible to identify and correct some errors and typos as well as improve the general level of English.

The changes made to the document are described below. All changes introduced and highlighted text segments appear in the final document highlighted in blue.

  • Line 1: delete “main” (OB also plays a part in VNO in some animals)

The word has been deleted, as proposed.

  • what are “neurogenerative diseases”?

This is an error, which has been replaced by "neurodegenerative diseases”, change that appears highlighted in blue.

  • The introduction contains many general well-known statements of trivial importance, which are in part disputable (e.g., the notion that dogs smell odorants at a much lower concentration than humans. This is strictly odorant-dependent and does not apply for all odorants (see https://pubmed.ncbi.nlm.nih.gov/24278296/)

Thus, the term “microsmatic or macrosmatic”), as often used in popular scientific publications, should be, if not avoided, at least explained.

The authors agree that the terminology "macrosmatic" and "microsmatic" are not unambiguously accepted, and as such the term has been removed from the text.

  • There are some unnecessary repetitions, e.g., the location of the OB.

Although throughout the text the location of the OB is referred to several times, in the different passages new information is added that, in general, allows a more accurate understanding of the location of this structure, namely:

“It is known that the OB, located in the rhinencephalon…”

“The OB is a bulbar structure located in the rostral extremity of each cerebral hemisphere, ventrocranially to the frontal lobe (25) and near the orbital portion of the frontal bone, arranged on each of the ethmoidal fossae of the cribriform plate through which it receives the constituent filaments of the olfactory nerve.”

“Within the cranial cavity the OB is located in the rostral cranial fossa,…”

  • Citations on the structure/function of neuroepithelium most often derive from mice and do not compare the situation in companion animals.

The authors agree that a significant part of the anatomical description of the nasal cavity and structures related to the olfactory system are based on information from rodents. This is due precisely to the fact that these species are much more explored and are better characterized from the histological and anatomical point of view than companion animals, which makes finding unequivocal information about these species more difficult. Even so, the authors considered that the information based on rodents do not vary greatly from those extrapolated to carnivores, allowing the reader to perceive the complexity of the anatomophysiological organization in these species, with the appropriate adaptations. Furthermore, a text segment was modified and enriched by introducing more specific information and references about the olfactory mucosa and neuroepithelium of dogs and cats, appearing highlighted in blue:

“In all domestic mammals, this olfactory mucosa is easily recognizable in vivo due to its brownish-yellowish coloring, which allows it to be distinguished from the adjacent pink respiratory mucosa (25, 39, 42). Airflow patterns through the nasal cavity in dogs are highly specialized, with the presence of a dorsal nasal meatus well separated from the respiratory airways and communicating directly with the well-developed olfactory recess. (43). This specialized separation of respiratory and olfactory air flows is also present in cats (39). In the olfactory recess, the odorants connect to olfactory receptor proteins on the cilia of olfactory sensory neurons arranged on the surface of the olfactory neuroepithelium. In dogs, this neuroepithelium is found covering not only the lamina cribrosa but also the caudodorsal region of the nasal septum, portions of the nasoturbinates and maxilloturbinates and of the ethmoturbinates (11, 41), and can contain between 6 and 10 million neurons (44) which are first-order neurons. In felines, the olfactory epithelium appears more concentrated in the medial regions of the olfactory recess and with a reduced peripheral distribution, which translates into a smaller number of olfactory sensory neurons and potentially contributes to a lower olfactory capacity of these animals (39, 45, 46).”

  • Are there any behavioral studies conducted that help in clinical diagnosis of animals? If so, please specify.

Reviewer 3 Report

The authors R.D. Alvites, A. Caine, G.B. Cherubini, J. Prada, A.S.P. Varejão and A.C. Maurício in theior contribution The Olfactory Bulb in companion animals - Anatomy, Physiology and Clinical Importance” review the morphofunctional significance of the olfactory bulb in companion animals, emphasizing its significance as a potential early indicator of pathologies and the significance of imaging evaluation techniques.

The paper highlights the importance of compiling existing studies on this topic, as it is an area with limited information but increasing clinical relevance, that has not been thoroughly explored, particularly in veterinary medicine.

Overall, the paper is well-organized and easy to read, with abundant information and valuable insights. The authors have also enriched the paper with their own histological and RM images.

However, there are some issues that need to be addressed. Firstly, while the literature review is comprehensive, there are some inconsistencies between the references used and the claims made in the text, which can be confusing for the reader. I have listed the most important ones, and I recommend that the authors revise this aspect.

Additionally, two of the figures are overlapping, making it difficult to view one of them properly. There are also some minor errors, which have been highlighted at the end of the paper.

In conclusion, this paper provides a valuable review of the literature on the olfactory bulb in domestic animals. However, to improve the quality and accuracy of the paper, I strongly suggest that the authors revise the inconsistencies in the references and correct the issues with the figures and minor errors. 

Misleading references in the bibliography

Although the bibliographic review is exhaustive, I have noticed that some of the citations in the text do not appear to directly support the points being made. It would be helpful to clarify the connection between the cited sources and the arguments presented.

- Line 46-50 in a general statement about the function of the olfactory bulb, this is supported by a very specific citation corresponding to a comparative morphological study between the OB of the dog and the goat in which the function of the OB is not the focus of the paper.

It seems appropriate to include a citation from a broader, more general scope such as “Shepherd, G. M., & Greer, C. A. (1998). Olfactory bulb. In G. M. Shepherd (Ed.), The synaptic organization of the brain (pp. 159-203). Oxford University Press.”

- Line 61: Reference [3] does not illustrate at all the evolutive history of the OB. It pertains to the investigation of how the morphology of the OB can be utilized as a biomarker for olfactory dysfunction. A more appropriate choice would be, for example: Poncelet G, Shimeld SM. The evolutionary origins of the vertebrate olfactory system. Open Biol. 2020 Dec;10(12):200330. doi: 10.1098/rsob.200330. Epub 2020 Dec 23. PMID: 33352063; PMCID: PMC7776563.

- Line 241-243: “Although some recent works have demonstrated that the two olfactory systems are essential for a correct interpretation of social and predation olfactory stimuli and are somewhat interrelated (12)”

The reference (12) “Dealing With Stress in Cats: What Is New About the Olfactory Strategy?” is not related with the interrelation between the Main and accessory olfactory system.

A more appropriated citation would be: “Suárez R, García-González D, de Castro F. Mutual influences between the main olfactory and vomeronasal systems in development and evolution. Front Neuroanat. 2012 Dec 24;6:50. doi: 10.3389/fnana.2012.00050. PMID: 23269914; PMCID: PMC3529325.”

- Line 274-276: “Unlike humans, who may have a few dozen receptors in each cilium, dogs can have more than 200 million receptors in the nasal cavity, maximizing their ability to identify a high number of odorants and in lower concentrations (34).”

Reference 34 is not appropriate here, as it is concerned with the accessory olfactory system, not the main olfactory system.

- Lines 388-391: “Similar alterations can be identified in older dogs (43), which show both a significant decrease in the number of cilia and olfactory receptors in the olfactory neuroepithelium and senile changes in OB such as cerebrovascular amyloidosis (34).”

Both are relevant assertions, but regretably the references are misleading. (43) is focused in fish and (34) is a review on the AOB that does not deal with pathologies, much certainly not in the MOB.

Other issues:

1) Line 26: “It is in this element that the first selective filtration of olfactory stimuli occurs through a complex cell interaction that forward the received olfactory information to higher cortical centers”.

Filtration refers to a mere selection of stimuli but actually the OB performs an integrative function of multiple simultaneous stimuli. I would therefore speak of “first selective integration”.

2) Line 50-52: “The capture, recognition, and processing of odorant molecules results from an interaction between the main and accessory olfactory systems.”

It is not until line 222 that the authors address the accessory olfactory system, so this early and decontextualized reference is confusing and misplaced.

3) Line 61-62: “Although neurogenesis in the adult brain only registers in the hippocampus and subventricular zone”.

It is more precise to say “mainly” instead of “only”. Jurkowski MP, Bettio L, K Woo E, Patten A, Yau SY, Gil-Mohapel J. Beyond the Hippocampus and the SVZ: Adult Neurogenesis Throughout the Brain. Front Cell Neurosci. 2020 Sep 29;14:576444. doi: 10.3389/fncel.2020.576444. PMID: 33132848; PMCID: PMC7550688.

4) Line 60-61: “The OB is one of the first structures to evolutionarily evolve in the vertebrate brain”

evolutionarily evolve is redundant; better evolutionarily appear.

5) Line 169-170: “The external plexiform layer contains the nuclei of tufted cells”

Somata instead of nuclei

6) Line 226-230: “The main olfactory system is concerned with detecting a large array of volatile odorants that translate into a consistent response known as a smell. The main olfactory system is the chemosensory system responsible for translating the information contained in an odorant and its awareness and perception as a smell.”

Both sentences are redundant.

7) Line 274-276: “Unlike humans, who may have a few dozen receptors in each cilium, dogs can have more than 200 million receptors in the nasal cavity, maximizing their ability to identify a high number of odorants and in lower concentrations (34).”

The fact that “humans may have a few dozen receptors in each cilium” should be supported with a reference.

Moreover, the comparison between the number of receptors in a single cilium for humans and the total number of receptors in a dog's nasal cavity is not a valid comparison, as these are distinct parameters. It is more appropriate to compare the total number of receptors in the nasal cavity for humans and dogs.

8) Line 290-294: The description of the olfactory nerve should be based in a reference, for instance, this: “Crespo C, Liberia T, Blasco-Ibáñez JM, Nácher J, Varea E. Cranial Pair I: The Olfactory Nerve. Anat Rec (Hoboken). 2019 Mar;302(3):405-427. doi: 10.1002/ar.23816. Epub 2018 Apr 23. PMID: 29659152.”

9) Line 410: “Currently there are no studies suggestin” suggesting

10) Line 417: “Recontration”. Do you mean reconstruction?

11) Figure 4 (Legend): “White arrows on each image highlight the thin curved lamina cribrosa (cribriform plate) that bounds the rostral aspect of the OBs, with multiple tiny defects through which the olfactory nerve bundles pass are visible in all planes although most distinct on the transverse plane compared to the reconstructions.” Meaningless sentence.

12) Figure 5 (legend): “Transverse slices from patients with this conformation therefore show only the OBs without any other portions of the brain.”

To be precise these transverse slices show dorsally to the OB more anterior part of the frontal lobe.

13) Line 436-438: “there are methods of volumetric determination by coronal T2-weighted segmentation applied in human medicine (60), although they currently have little clinical relevance”

“It is somewhat puzzling to read that these methods have little clinical relevance when earlier in the text mention has been made of several papers linking a reduction in OB volume in humans with various pathologies.”

14) Line 455: “While some studies indicate otherwise (63)”

It would be interesting to add one more reference that support this assertion, in this case from a morphological point of view: “Bird DJ, Jacquemetton C, Buelow SA, Evans AW, Van Valkenburgh B. Domesticating olfaction: Dog breeds, including scent hounds, have reduced cribriform plate morphology relative to wolves. Anat Rec (Hoboken). 2021 Jan;304(1):139-153. doi: 10.1002/ar.24518. Epub 2020 Nov 17. PMID: 33205623.”

15) Figure 6: Letter e is missing.

16) Line 588: Wei, Q., Zhang, H., & Guo, B. (2008). Histological Structure Difference of Dog's Olfactory Bulb Between Different Age and Sex. 

The information regarding the Journal, volume, pages is missing [Zoological Research, 29, 537-545]

17) Figure 2 is hidden behind Figure 1, so it cannot be evaluated.

18) Have the authors considered the interest of these references?

Prichard A, Chhibber R, King J, Athanassiades K, Spivak M, Berns GS. Decoding Odor Mixtures in the Dog Brain: An Awake fMRI Study. Chem Senses. 2020 Dec 5;45(9):833-844. doi: 10.1093/chemse/bjaa068. PMID: 33179730.

Jacquemetton C, Drexler A, Kellerman G, Bird D, Van Valkenburgh B. The impact of extreme skull morphology in domestic dogs on cribriform plate shape. Anat Rec (Hoboken). 2021 Jan;304(1):190-201. doi: 10.1002/ar.24512. Epub 2020 Sep 30. PMID: 33000502.

Throughout the work there are some typos and misspellings that should be corrected. I include in my review some of them. 

Author Response

Answer to reviewer 3

Dear reviewer 3:

Thank you very much for the feedback on this review phase, and also for the suggestions made, which received the best attention from us. The authors inform that the final document has been revised and all suggestions made by the reviewers have been introduced, being duly identified in the final document with highlight in different colors. This review also made it possible to identify and correct some errors and typos as well as improve the general level of English.

The changes made to the document are described below. All changes introduced and highlighted text segments appear in the final document highlighted in yellow.

  • Line 46-50 in a general statement about the function of the olfactory bulb, this is supported by a very specific citation corresponding to a comparative morphological study between the OB of the dog and the goat in which the function of the OB is not the focus of the paper.

It seems appropriate to include a citation from a broader, more general scope such as “Shepherd, G. M., & Greer, C. A. (1998).Olfactory bulb. In G. M. Shepherd (Ed.), The synaptic organization of the brain (pp. 159-203). Oxford University Press.”

The proposed reference has been introduced into the document.

  • Line 61: Reference [3] does not illustrate at all the evolutive history of the OB. It pertains to the investigation of how the morphology of the OB can be utilized as a biomarker for olfactory dysfunction. A more appropriate choice would be, for example: Poncelet G, Shimeld SM. The evolutionary origins of the vertebrate olfactory system. Open Biol. 2020 Dec;10(12):200330. doi: 10.1098/rsob.200330. Epub 2020 Dec 23. PMID: 33352063; PMCID: PMC7776563.

The proposed reference has been introduced into the document. In fact the reference had already been used, but in another part of the article.

  • Line 241-243: “Although some recent works have demonstrated that the two olfactory systems are essential for a correct interpretation of social and predation olfactory stimuli and are somewhat interrelated (12)”

The reference (12) “Dealing With Stress in Cats: What Is New About the Olfactory Strategy?” is not related with the interrelation between the Main and accessory olfactory system.

A more appropriated citation would be: “Suárez R, García-González D, de Castro F. Mutual influences between the main olfactory and vomeronasal systems in development and evolution. Front Neuroanat. 2012 Dec 24;6:50. doi: 10.3389/fnana.2012.00050. PMID: 23269914; PMCID: PMC3529325.”

The proposed reference has been introduced into the document.

  • Line 274-276: “Unlike humans, who may have a few dozen receptors in each cilium, dogs can have more than 200 million receptors in the nasal cavity, maximizing their ability to identify a high number of odorants and in lower concentrations (34).” Reference 34 is not appropriate here, as it is concerned with the accessory olfactory system, not the main olfactory system.

A new and more adequate reference has been introduced into the document:

Jenkins, E. K., DeChant, M. T., & Perry, E. B. (2018). When the nose doesn’t know: Canine olfactory function associated with health, management, and potential links to microbiota. Frontiers in veterinary science, 56.”

  • Lines 388-391: “Similar alterations can be identified in older dogs (43), which show both a significant decrease in the number of cilia and olfactory receptors in the olfactory neuroepithelium and senile changes in OB such as cerebrovascular amyloidosis (34).”

Both are relevant assertions, but regretably the references are misleading. (43) is focused in fish and (34) is a review on the AOB that does not deal with pathologies, much certainly not in the MOB.

The two references were compiled into one more suited to the text segment.

“Hirai, T., Kojima, S., Shimada, A., Umemura, T., Sakai, M., & Itakurat, C. (1996). Age‐related changes in the olfactory system of dogs. Neuropathology and applied neurobiology, 22(6), 531-539.”

  • Line 26: “It is in this element that the first selective filtration of olfactory stimuli occurs through a complex cell interaction that forward the received olfactory information to higher cortical centers”.
  • Filtration refers to a mere selection of stimuli but actually the OB performs an integrative function of multiple simultaneous stimuli. I would therefore speak of “first selective integration”.

The suggested change was introduced in the abstract, with the change appearing highlighted in yellow:

It is in this element that the first selective integration of olfactory stimuli occurs through a complex cell interaction that forward the received olfactory information to higher cortical centers.”

  • Line 50-52: “The capture, recognition, and processing ofodorant molecules results from an interaction between the mainand accessory olfactory systems.”

It is not until line 222 that the authors address the accessoryolfactory system, so this early and decontextualized reference isconfusing and misplaced.

“The textual segment has been modified, with the change appearing highlighted in yellow:

“The capture, recognition, and processing of odorant molecules results from an interaction between different components of the olfactory system.”

  • Line 61-62: “ Although neurogenesis in the adult brain only registers in the hippocampus and subventricular zone”. It is more precise to say “mainly” instead of “only”. Jurkowski MP,Bettio L, K Woo E, Patten A, Yau SY, Gil-Mohapel J. Beyond the Hippocampus and the SVZ: Adult Neurogenesis Throughout the Brain. Front Cell Neurosci. 2020 Sep 29;14:576444. doi:10.3389/fncel.2020.576444. PMID: 33132848; PMCID:PMC7550688.

The amendment and the proposed reference have been introduced into the text, with the change appearing highlighted in yellow:

“although neurogenesis in the adult brain registers mainly in the hippocampus and sub-ventricular zone…”

  • Line 60-61: “The OB is one of the first structures to evolutionarily evolve in the vertebrate brain” evolutionarily evolve is redundant; better evolutionarily appear

The amendment have been introduced into the text, with the change appearing highlighted in yellow:

“The OB is one of the first structures to evolutionarily appear  in the vertebrate brain.”

  • Line 169-170: “The external plexiform layer contains the nuclei of tufted cells”

Somata instead of nuclei

The amendment have been introduced into the text, with the change appearing highlighted in yellow:

“The external plexiform layer contains the somata of tufted cells”

  • Line 226-230: “ The main olfactory system is concerned with detecting a large array of volatile odorants that translate into a consistent response known as a smell. The main olfactory system is the chemosensory system responsible for translating the information contained in an odorant and its awareness and perception as a smell.”

Both sentences are redundant.

The second sentence was eliminated to avoid redundancy.

  • Line 274-276: “Unlike humans, who may have a few dozen receptors in each cilium, dogs can have more than 200 million receptors in the nasal cavity, maximizing their ability to identify ahigh number of odorants and in lower concentrations (34).” The fact that “humans may have a few dozen receptors in each cilium” should be supported with a reference. Moreover, the comparison between the number of receptors in a single cilium for humans and the total number of receptors in a dog's nasal cavity is not a valid comparison, as these are distinct parameters. It is more appropriate to compare the total number of receptors in the nasal cavity for humans and dogs.

The passage has been slightly modified and a new reference has been introduced, with the change appearing highlighted in yellow:

“Unlike humans, who may have around 50 million receptors (44), dogs can have more than 200 million olfactory receptors in the nasal cavity, maximizing their ability to identify a high number of odorants and in lower concentrations”

“Sarafoleanu, C., Mella, C., Georgescu, M., & Perederco, C. (2009). The importance of the olfactory sense in the human behavior and evolution. Journal of Medicine and life, 2(2), 196.”

  • Line 290-294: The description of the olfactory nerve should be based in a reference, for instance, this: “Crespo C, Liberia T,Blasco-Ibáñez JM, Nácher J, Varea E. Cranial Pair I: The Olfactory Nerve. Anat Rec (Hoboken). 2019 Mar;302(3):405-427.doi: 10.1002/ar.23816. Epub 2018 Apr 23. PMID: 29659152.”

The proposed reference has been introduced.

  • Line 410: “Currently there are no studies suggestin ” suggesting.

The amendment have been introduced into the text, with the change appearing highlighted in yellow:

“Currently there are no studies suggesting…”

  • Line 417: “Recontration”. Do you mean reconstruction?

Yes. The error has been corrected.

“…with small perforations visible on high resolution reconstructions…”

  • Figure 4 (Legend): “ White arrows on each image highlight the thin curved lamina cribrosa (cribriform plate) that bounds the rostral aspect of the OBs, with multiple tiny defects through which the olfactory nerve bundles pass are visible in all planes although most distinct on the transverse plane compared to there constructions.” Meaningless sentence.

The passage has been slightly modified to enhance the intended message, with the change appearing highlighted in yellow:

“White arrows on each image highlight the thin curved lamina cribrosa (cribriform plate) that bounds the rostral aspect of the OBs. Multiple tiny defects through which the olfactory nerve bundles pass are visible in all planes, although most distinct on the transverse plane compared to the remaining ones.”

  • Figure 5 (legend): “Transverse slices from patients with this conformation therefore show only the OBs without any other portions of the brain.”

The passage has been slightly modified, with the change appearing highlighted in yellow:

“Transverse slices from patients with this conformation therefore show only the OBs and small segments of the surrounding brain.”

  • 13) Line 436-438: “there are methods of volumetric determination by coronal T2-weighted segmentation applied in human medicine (60), although they currently have little clinical relevance”

“It is somewhat puzzling to read that these methods have little clinical relevance when earlier in the text mention has been made of several papers linking a reduction in OB volume in humans with various pathologies.”

The passage has been slightly modified to enhance the intended message, with the change appearing highlighted in yellow:

“… although the use of these techniques in routine clinical practice is still limited.”

  • Line 455: “While some studies indicate otherwise (63)”

It would be interesting to add one more reference that support this assertion, in this case from a morphological point of view:“ Bird DJ, Jacquemetton C, Buelow SA, Evans AW, VanValkenburgh B. Domesticating olfaction: Dog breeds, includingscent hounds, have reduced cribriform plate morphology relativeto wolves. Anat Rec (Hoboken). 2021 Jan;304(1):139-153. doi:10.1002/ar.24518. Epub 2020 Nov 17. PMID: 33205623.”

The proposed reference was introduced.

  • Figure 6: Letter e is missing.

The missing letter has been introduced.

  • Line 588: Wei, Q., Zhang, H., & Guo, B. (2008). Histological Structure Difference of Dog's Olfactory Bulb Between DifferentAge and Sex. The information regarding the Journal, volume, pages is missing[Zoological Research, 29, 537-545].

Reference data has been updated.

  • Figure 2 is hidden behind Figure 1, so it cannot be evaluated.

The figures must have appeared overlapping due to an error in the version of the document received by the reviewer. Below are the two individualized images and captions:

  • Have the authors considered the interest of these references?

Prichard A, Chhibber R, King J, Athanassiades K, Spivak M,Berns GS. Decoding Odor Mixtures in the Dog Brain: An AwakefMRI Study.

Chem Senses. 2020 Dec 5;45(9):833-844. doi:10.1093/chemse/bjaa068. PMID: 33179730.

New information has been added to accommodate the proposed reference, with the change appearing highlighted in yellow:

“In addition to the white matter tracts connecting the OB and the piriform cortex, the limbic system and the entorhinal cortex, tracts connecting the OB and the occipital corte, the posterior cingulate…”

  • jacquemetton C, Drexler A, Kellerman G, Bird D, VanValkenburgh B. The impact of extreme skull morphology indomestic dogs on cribriform plate shape. Anat Rec (Hoboken). 2021 Jan;304(1):190-201. doi: 10.1002/ar.24512. Epub 2020Sep 30. PMID: 33000502.

A small new passage has been introduced for the introduction of the proposed reference, with the change appearing highlighted in yellow:

“There even seems to be a relationship between the morphological plasticity of the lamina cribrosa and the pressure induced by artificial selection in domestic canids compared to wild species.”

  • Throughout the work there are some typos and misspellings that should be corrected. I include in my review some of them.

A careful review was carried out throughout the document to correct minor errors identified and improve English.

Round 2

Reviewer 2 Report

issues are addressed

Reviewer 3 Report

The authors have made detailed corrections and responses to my suggestions. They have done so to a satisfactory level, and as a result, the quality of the article has significantly increased.